# Emulating Clinical Diagnostic Reasoning for Jaw Cysts with Machine Learning

**DOI:** 10.3390/diagnostics12081968

**Published:** 2022-08-14

**Authors:** Balazs Feher, Ulrike Kuchler, Falk Schwendicke, Lisa Schneider, Jose Eduardo Cejudo Grano de Oro, Tong Xi, Shankeeth Vinayahalingam, Tzu-Ming Harry Hsu, Janet Brinz, Akhilanand Chaurasia, Kunaal Dhingra, Robert Andre Gaudin, Hossein Mohammad-Rahimi, Nielsen Pereira, Francesc Perez-Pastor, Olga Tryfonos, Sergio E. Uribe, Marcel Hanisch, Joachim Krois

**Affiliations:** 1Department of Oral Surgery, University Clinic of Dentistry, Medical University of Vienna, 1090 Vienna, Austria; 2Competence Center Oral Biology, University Clinic of Dentistry, Medical University of Vienna, 1090 Vienna, Austria; 3Department of Oral Diagnostics, Digital Health, and Health Services Research, Charité—University Medicine Berlin, 14197 Berlin, Germany; 4Department of Oral and Maxillofacial Surgery, Radboud University Nijmegen Medical Centre, 6525 GA Nijmegen, The Netherlands; 5Computer Science and Artificial Intelligence Laboratory, Massachusetts Institute of Technology, Cambridge, MA 02139, USA; 6Department of Restorative Dentistry, Ludwig-Maximilians-University of Munich, 80336 Munich, Germany; 7Department of Oral Medicine and Radiology, Faculty of Dental Sciences, King George’s Medical University, Lucknow 226003, India; 8Periodontics Division, Centre for Dental Education and Research, All India Institute of Medical Sciences, New Delhi 110029, India; 9Department of Oral and Maxillofacial Surgery, Charité—University Medicine Berlin, 14197 Berlin, Germany; 10Berlin Institute of Health, 10178 Berlin, Germany; 11Dentofacial Deformities Research Center, Research Institute of Dental Sciences, Shahid Beheshti University of Medical Sciences, Tehran 1416634793, Iran; 12Private Practice in Oral and Maxillofacial Radiology, Rio de Janeiro 22430-000, Brazil; 13Servei Salut Dental, Gerencia Atencio Primaria, Institut Balear de la Salut, 07003 Palma, Spain; 14Department of Periodontology and Oral Biochemistry, Academic Centre for Dentistry Amsterdam, 1081 LA Amsterdam, The Netherlands; 15Department of Conservative Dentistry & Oral Health, Riga Stradins University, LV-1007 Riga, Latvia; 16School of Dentistry, Universidad Austral de Chile, Valdivia 5110566, Chile; 17Baltic Biomaterials Centre of Excellence, Headquarters at Riga Technical University, LV-1658 Riga, Latvia; 18Department of Oral and Maxillofacial Surgery, University Clinic Münster, 48143 Münster, Germany

**Keywords:** artificial intelligence, machine learning, surgery, oral, radiography, cysts, diagnosis

## Abstract

The detection and classification of cystic lesions of the jaw is of high clinical relevance and represents a topic of interest in medical artificial intelligence research. The human clinical diagnostic reasoning process uses contextual information, including the spatial relation of the detected lesion to other anatomical structures, to establish a preliminary classification. Here, we aimed to emulate clinical diagnostic reasoning step by step by using a combined object detection and image segmentation approach on panoramic radiographs (OPGs). We used a multicenter training dataset of 855 OPGs (all positives) and an evaluation set of 384 OPGs (240 negatives). We further compared our models to an international human control group of ten dental professionals from seven countries. The object detection model achieved an average precision of 0.42 (intersection over union (IoU): 0.50, maximal detections: 100) and an average recall of 0.394 (IoU: 0.50–0.95, maximal detections: 100). The classification model achieved a sensitivity of 0.84 for odontogenic cysts and 0.56 for non-odontogenic cysts as well as a specificity of 0.59 for odontogenic cysts and 0.84 for non-odontogenic cysts (IoU: 0.30). The human control group achieved a sensitivity of 0.70 for odontogenic cysts, 0.44 for non-odontogenic cysts, and 0.56 for OPGs without cysts as well as a specificity of 0.62 for odontogenic cysts, 0.95 for non-odontogenic cysts, and 0.76 for OPGs without cysts. Taken together, our results show that a combined object detection and image segmentation approach is feasible in emulating the human clinical diagnostic reasoning process in classifying cystic lesions of the jaw.

## 1. Introduction

Jaw cysts are highly prevalent [1] yet frequently asymptomatic; thus, they often remain undiagnosed until their dimensions require radical surgery [2,3]. At such a late stage, the extent of the defect can present a risk for neighboring anatomical structures, including teeth, alveolar bone, and nerves [4]. Furthermore, the time required for complete postoperative osseous regeneration is exponentially proportional to the preoperative volume of the defect [5]. Consequently, timely diagnosis ensures a smaller osseous defect and a shorter regeneration time, thus an overall better prognosis.

While jaw cysts are identifiable at an early stage on panoramic radiographs (i.e., orthopantomograms, OPGs) [1], in practice, they are usually incidental findings [6]. Supporting the radiological diagnosis of jaw cysts has thus been a focus of artificial intelligence research in oral medicine [7,8,9]. Specifically, previous work has applied both object detection [10] and classification [11] methods to oral cyst diagnostics using OPGs. Notwithstanding, the explainability of existing methods remains a concern [1,12,13], and no previous work has focused on a machine learning approach that purposefully emulates the explainable human thought process in a clinical setting.

Clinically, experience from past encounters as well as contextual knowledge, including the spatial relation of the cystic lesion to neighboring anatomical structures (e.g., proximity of a tooth apex to a radicular cyst) is frequently used to establish a preliminary diagnosis until further imaging is performed [14,15] or a definitive histopathological diagnosis is made. This thought process is well-described in medical research and education as clinical diagnostic reasoning [16,17,18].

We hypothesized that clinical diagnostic reasoning can be emulated by using machine learning to individually replicate each step: first detecting a cystic lesion, then recognizing neighboring anatomical structures and their proximity to or overlap with the lesion, and finally using these as contextual information to establish a preliminary classification. Thus, the aim of our study was to purposefully emulate the human clinical diagnostic reasoning process step by step through the implementation of a combined object detection and image segmentation approach for the detection and preliminary classification of cystic lesions on OPGs.

## 2. Materials and Methods

### 2.1. Image Data

#### 2.1.1. Data Collection

We collected OPGs with cystic lesions of the jaw at the University Clinic of Dentistry of the Medical University of Vienna between 2000 and 2020 as well as at the Radboud University Nijmegen Medical Centre between 2012 and 2019. To be included in the training dataset, the cystic lesions on the OPGs had to have a confirmed histopathological diagnosis. We further randomly collected negative (i.e., no cystic lesions of the jaw) OPGs from the same period to achieve a clinically representative positive–negative ratio in the test set. We aimed to achieve a comparable patient age and sex distribution between positive and negative OPGs (Table 1).

#### 2.1.2. Data Preparation

OPGs at the Medical University of Vienna were taken with an Ortophos SL or an Orthophos XG Plus device (Dentsply Sirona, York, PA, USA) at 70 kV and 15 mA. OPGs at the Radboud University Nijmegen Medical Centre were taken with a Cranex Novus e device (Soredex, Helsinki, Finland) at 77 kV and 10 mA, using a CCD sensor. For data preparation, we downloaded all OPGs as DICOM files from the respective imaging databases and de-identified them. Next, we labeled ground truth OPGs by recording the (x,y) coordinates of the upper-left corner as well as the width and height of the rectangle with the smallest area containing the radiolucent lesion (i.e., ground truth bounding box) into a separate text file. We then added age, sex, as well as the confirmed histopathological diagnosis to the file. The training dataset consisted of a total of 855 OPGs containing cysts; all eligible OPGs within the respective time frames were included in the training dataset, and no sample size calculation was performed. Negative OPGs were not included in the training dataset, as those do not contribute to the learning. The test set had 384 OPGs in total, of which 240 were negative, yielding a prevalence of approximately 38% to measure the performance of the model on a clinically representative sample.

### 2.2. Modeling

We employed a combined object detection and image segmentation model for the detection and preliminary classification of cysts. The overview of the architecture is represented in Figure 1. It consists of three major elements: object detection model, segmentation model, and Random Forest classifier. The object detection model detects the location of the cyst. Simultaneously, multiple segmentation models segment anatomical structures of relevance. Detection boxes and segmented anatomical structures are then combined, and their overlaps are computed. Finally, a Random Forest classifier classifies the cysts based on the overlap values.

#### 2.2.1. Object Detection Modeling

Here, we used RetinaNet as our object detection model. RetinaNet is a single-stage algorithm that combines Feature Pyramid Networks to address the problem of scale invariance [19] with a loss function (i.e., Focal loss [20]) to address the imbalance between foreground and background in the set of candidate object locations. We used a ResNet50 [21] pretrained on the ImageNet dataset as the backbone. The output of the model is a set of bounding boxes together with labels and confidence scores.

For model training, OPGs were kept at their original size and resolution. As OPGs are grayscale images, to feed them to the model, we converted them into RGB images by triplicating the single channel. We trained the model for a maximum of 20 epochs. The batch size was set to 8, and the Adam optimizer with a learning rate of 0.001 was used for training, as it adapts individual learning rates for each network parameter. Given the small size of our dataset, we decided to apply early stopping during training to avoid overfitting. We monitored the validation loss and stopped training if this loss did not improve after five epochs.

After training, we evaluated the model on the evaluation set. We compared the output of the model against our annotations using typical object detection metrics (e.g., mean average precision, recall). These metrics are calculated for different ranges of intersection over union (IoU) values, which show the level of overlap between the predicted and ground truth bounding boxes. In a second step, we translated the output of the model and the evaluation set into binary classes. We performed the evaluation for several threshold values for the confidence values of the predictions of the model, using accuracy, precision, sensitivity, specificity, and F1-score as metrics.

#### 2.2.2. Segmentation Models

Segmentation models, which perform a classification task at the pixel level, were used to segment relevant anatomical structures on the OPGs. We trained one model each for the maxilla, mandible, mandibular canal, maxillary sinuses, dentition, and individual teeth. For the sake of simplicity, each task was solved with the same U-Net++ model architecture, which was initialized with pretrained weights from ImageNet [22]. The basic model architecture was extended by the specialized layers of a VGG19 [23], which showed good performance for dental radiographic image analysis [24]. Training was performed over 200 epochs with the Adam optimizer with a learning rate of 0.0001 and a batch size of 8. The loss was constructed through the unweighted sum of Dice loss and Focal loss [20]. The data were split into train, validation, and test data with a proportion of 80%, 10% and 10%, respectively. The 80–10–10 split is a widely used selection, as it balances a good tradeoff for large datasets by providing enough data for training to achieve an accurate model while keeping enough data aside to compute a representative test performance. Training images were augmented with random Gaussian noise (*p* = 0.3) and random shifts in image intensity (*p* = 0.3). For the sake of simplicity, we kept hyperparameters consistent over all tasks and did not perform an extensive hyperparameter search for each task individually. All models were trained on one Quadro RTX 8000 with 48 GB of VRAM (Nvidia, Santa Clara, CA, USA).

#### 2.2.3. Mask Overlap

As discussed above, the overlay of cystic lesions with anatomical structures (e.g., teeth) may improve diagnostic performance. Hence, the masks generated from the different segmentation models were overlapped with the bounding box with the highest confidence returned by the object detection model. This results in a value between 0 and 1, which states the share of the bounding box that is covered by the specific segmentation mask. Overlay values were computed for segmentation masks of maxilla, mandible, mandibular canals, maxillary sinuses, dentition and one value each for individual teeth. The resulting values for each sample were used as input features for the next processing step.

#### 2.2.4. Random Forest Classifier

Random Forests are an ensemble machine learning model that uses a multitude of decision trees for training. The algorithm combines the idea of “bagging”, where random samples are repeatedly drawn with replacement with random feature selection of the training set [25]. This prevents overfitting and makes Random Forest models quite robust across different applications. As input features of the algorithm, we used the overlap ratio of a bounding box containing the cysts with the masks that show the individual anatomic structures (maxilla, mandible, mandibular canals, maxillary sinuses, dentition, and individual teeth). The model is trained to predict two classes: odontogenic and non-odontogenic cysts (Figure 1).

The data were split into 80% training and 20% test data. Training and testing were performed on input features of the overlaps of annotated bounding boxes and the generated segmentation masks. The hyperparameter tuning of the Random Forest was performed with a grid search, which exhaustively considers all parameter combinations defined. As a result, the Random Forest was trained with 1000 random decision trees, which performed decision splits based on the gini impurity as information gain. The trained Random Forest model is later used to predict the two classes, odontogenic and non-odontogenic cysts, for the overlaps of bounding boxes detected by the object detection model with the segmentation masks of the individual anatomic structures. Naturally, the test set used to evaluate the performance of the model consists of the same samples as used within this final prediction. This inference is performed multiple times to consider the variable threshold of the object detection model. Note that with higher thresholds, the number of remaining samples may shrink, as samples without a reported bounding box with a confidence above the threshold were removed.

### 2.3. Human Control Group

In order to evaluate the performance of our models in emulating the human clinical diagnostic reasoning process, we compared it to an international control group which consisted of ten dental professionals from seven different countries. The median experience of the human control group was 15 years (interquartile range (IQR): 8–27). Two members of the human control group (20%) were trained in oral and maxillofacial radiology. The human control group reviewed a batch of 301 randomly selected OPGs and determined whether the radiographs contained cystic lesions. Furthermore, upon the diagnosis of a cystic lesion, the human control group determined whether the cyst showed signs of odontogenic or non-odontogenic pathogenesis. Cystic lesions with an odontogenic pathogenesis had to show clear radiological signs associated with an underlying dental pathology (e.g., an apex protruding into a radicular cyst). Cystic lesions with a non-odontogenic pathogenesis had to show no radiologically discernible signs of an underlying dental pathology in their close proximity. Importantly, this only included causative dental factors, not symptoms potentially resulting from cystic growth (e.g., tooth displacement or resorption). Sample OPGs for odontogenic as well as non-odontogenic cysts are shown in Figure A1.

For analysis, we made the OPGs available to the human control group through an online platform (Mono, dentalXrai, Berlin, Germany) that allowed them to dynamically adjust brightness and contrast as well as to revise OPGs they already reviewed once. The human control group was trained on the definitions of the diagnoses as well as the usage of the platform before conducting their analysis. The inter-rater agreement is assessed using Fleiss’ Kappa [26], whereas the diagnostic performance of the individual dental professionals is assessed via sensitivity, specificity, precision (positive predictive value), negative predictive value, and the F1-score.

## 3. Results

### 3.1. Dataset Characteristics

In total, we included 1239 OPGs in this study. The Vienna subset included 790 positive OPGs (median patient age: 50 years, IQR: 39–60, 43% female, 57% male) and 120 negative OPGs (median patient age: 51 years, IQR: 43–55, 43% female, 57% male). The Nijmegen subset included 241 positive OPGs (median patient age: 57 years, IQR: 45–66, 35% female, 65% male) and 88 negative OPGs (median patient age: 51 years, IQR: 42–57, 36% female, 64% male). Complete characteristics of the dataset are shown in Table 1.

### 3.2. Detection and Segmentation Performance

The object detection model achieved an average precision of 0.42 (IoU: 0.50, maximal detections: 100) and an average recall of 0.39 (IoU: 0.50–0.95, maximal detections: 100). Complete metrics for the detection performance are shown in Table A1.

The segmentation models reached Dice scores of 0.945 (dentition), 0.794 (mandibular canal), 0.644 (maxillary sinus), 0.465 (maxilla), 0.978 (mandible) and 0.870 (individual tooth) on their unseen test sets.

### 3.3. Random Forest Classifier

The performances reached by the Random Forest classifiers are reported in Table 2. First, we report results from the Random Forest classifiers based on predicted cysts via object detection with an IoU over 0.3. The threshold was selected through a performance analysis over different thresholds as represented in Figure A2. For comparison, we also report results from the Random Forest classifiers based on the original annotations.

### 3.4. Human Diagnostic Performance

Finally, we compared the performance of our models against ten dental professionals who evaluated a dataset of 301 OPGs. This human control group had to determine whether an OPG contained a cystic lesion and if yes, whether the lesion was odontogenic or non-odontogenic in nature. Regarding odontogenic cysts, the human control group showed a sensitivity of 0.70 and a specificity of 0.62. Regarding non-odontogenic cysts, the human control group showed a sensitivity of 0.44 and a specificity of 0.95. Regarding healthy (i.e., no cystic lesions) patients, the human control group showed a sensitivity of 0.56 and a specificity of 0.76. The inter-rater agreement within the human control group was κ = 0.28 (95% confidence interval: 0.27–0.29). A summary of the human diagnostic performance is shown in Table 3, and a breakdown of each individual human reviewer is shown in Table A2. A comparison of diagnostic performance between human controls and classification models is shown in Figure 2.

## 4. Discussion

In this study, we aimed to apply a combined object detection and image segmentation approach to emulate clinical diagnostic reasoning in the detection and classification of cystic lesions on OPGs. Our object detection model achieved an average precision of 0.42 (IoU: 0.50, maximal detections: 100) and an average recall of 0.394 (IoU: 0.50–0.95, maximal detections: 100). Our classification model achieved a sensitivity of 0.84 for odontogenic cysts and 0.56 for non-odontogenic cysts as well as a specificity of 0.59 for odontogenic cysts and 0.84 for non-odontogenic cysts (IoU: 0.30). Comparing our results to an international human control group of ten dental professionals, we found that the human control group achieved a sensitivity of 0.70 for odontogenic cysts, 0.44 for non-odontogenic cysts, and 0.56 for OPGs without cysts as well as a specificity of 0.62 for odontogenic cysts, 0.95 for non-odontogenic cysts, and 0.76 for OPGs without cysts. Notwithstanding the variability inside the human control group, these results are largely comparable to the results from our classification model. Taken together, the results support the plausibility of our approach in emulating clinical diagnostic reasoning in detecting and classifying jaw cysts.

The novelty of our study lies in its aim as well as its use of both multicenter datasets and international human controls. As opposed to developing models with the highest detection accuracy, we specifically aimed to replicate the multi-step thought process of human clinical reasoning in the radiographic diagnosis of jaw cysts. While the simultaneous detection and classification of jaw cysts and tumors has previously been published with good results [27], to our knowledge, our combined object detection and image segmentation approach is the first that is deliberately analogous to the way a clinician makes a preliminary diagnosis. Furthermore, to mitigate location bias, we used a multicenter dataset to train our models and then compared them to an international human control group which consisted of dental professionals from seven different countries with different dental backgrounds and education levels. The size of our datasets is largely comparable to the most recent work in this field [10,11]. Notably, one recent study used more than eleven times as many negatives as positives for pretraining, resulting in a massive overall dataset from a single center [1]. Nonetheless, the number of positives is comparable to our multicenter dataset. In contrast to our work, this previous study also applied segmentation masks to the lesions themselves. While segmentation masks can be more accurate than bounding boxes, cystic lesions do not always present as sharply defined radiolucencies on OPGs, hence our decision to use bounding boxes.

Regarding the individual models in our study, it is apparent that the classification model performs better than the object detection model. In fact, the difference in F1-score between the classification models based on predicted and ground truth bounding boxes was only 0.01 for odontogenic cysts and 0.08 for non-odontogenic cysts (IoU: 0.30). This implies a good differentiability of odontogenic and non-odontogenic cysts based on their spatial relations to neighboring anatomical structures. Importantly, this differentiability does not seem to suffer from imperfect detection of the lesions themselves. A further performance difference can be observed between the classification of odontogenic and non-odontogenic cysts. A plausible explanation is the lower heterogeneity with which odontogenic cysts appear on OPGs. Odontogenic cysts present as periapical radiolucencies; thus, any detected odontogenic cyst shows a high overlap with the segmentation model for individual teeth. Such strict anatomical requirements are not applicable for non-odontogenic cysts, which might explain the lower classification performance of our models. It should also be noted that our maxillary segmentation model reached a lower Dice score (0.465) compared with our other segmentation models (0.644–0.978). One possible explanation is the extensive overlap of the maxilla with other anatomical structures in OPGs. The upper dentition, particularly in the posterior, nearly completely overlaps the maxilla itself, rendering recognition potentially difficult. The maxilla further does not feature any large, distinct, overlap-free area. In comparison, the mandibular rami are substantial, easily recognizable areas with almost no overlap at all except for the mandibular foramen.

Our study is limited by the low number of study centers which potentially compromised the generalizability of the results. We further did not differentiate between histopathological diagnoses other than odontogenic and non-odontogenic cysts. While this was a deliberate decision to lower class imbalance and increase the number of OPGs per diagnosis, a more granular differentiation would have allowed the development of a classifier with higher clinical applicability. To do so, a larger sample size would have been needed, especially for histopathological diagnoses with lower prevalence (e.g., ameloblastoma). This would also enable screening for lesions with aggressive growth and potential malignant transformations (e.g., keratocyst) [28]. A limitation with regard to the human control group is that while its members were trained on the definitions of the diagnoses as well as the usage of the online platform before conducting their analysis, no further calibration was performed. This, along with their varying experience levels, represents a potential source of bias in the results of the human control group.

The clinical implications of our results are twofold. First, our models could be utilized to aid diagnostics as well as the surgical decision-making process. Several deep learning tools are already applied in clinical practice [29,30], yet to our knowledge, this is the first study augmenting the object detection of cystic lesions with an image segmentation approach of neighboring anatomical structures to predict the pathogenesis of the lesion. This methodology mimics the human clinical decision-making process in everyday practice. In combination with a clinical examination, our classifier can be used to determine whether a dental pathology is involved, which further influences the treatment. The necessity of a thorough clinical examination should be emphasized, as our models do not provide information with regard to the vitality, mobility, and discoloration of affected or neighboring teeth. Thus, in clinical practice, our models should only be used in combination with a clinical examination. Second, our approach could serve as a baseline for further research into machine learning methods in the diagnosis of cysts and tumors of the jaw. Recent work has already employed radiomics to identify features characteristic to certain lesions. This could be combined with our analysis of spatial relations of the lesions to neighboring anatomical structures to increase diagnostic accuracy potentially further.

## 5. Conclusions

Within the limitations of the study, our results show that a combined object detection and image segmentation approach is feasible in emulating clinical diagnostic reasoning to classify cystic lesions of the jaw.

## Figures and Tables

**Figure 1 diagnostics-12-01968-f001:**
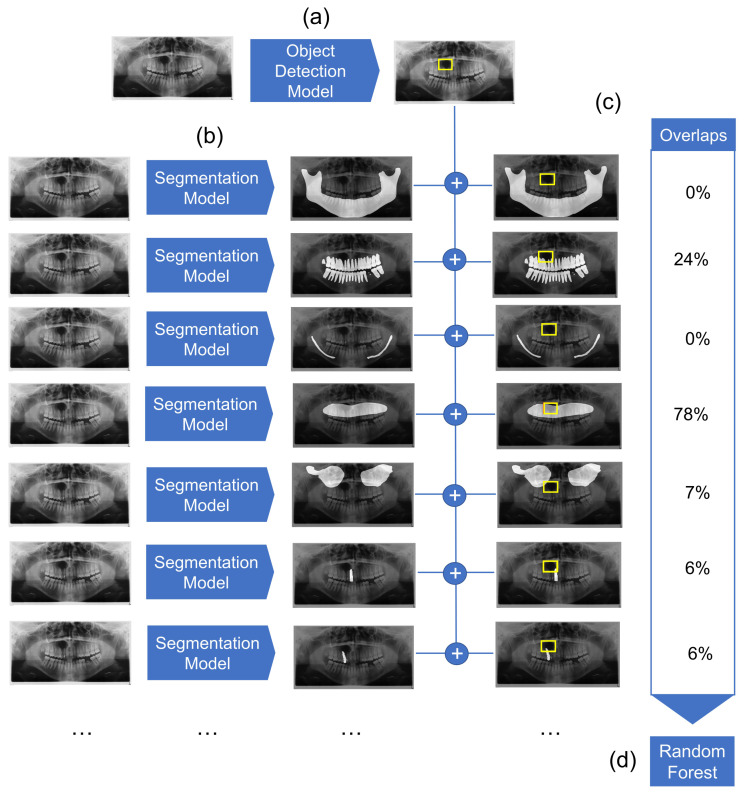
Overview of the diagnostic pipeline. (**a**) First, the object detection model detects cystic lesions and marks them with a bounding box. (**b**) Next, multiple segmentation models are employed which segment structures of maxilla, mandible, mandibular canal, maxillary sinuses, the complete dentition, as well as each individual tooth. (**c**) The overlaps between the marked bounding box and segmented structures are then calculated. (**d**) Finally, the Random Forest classifier gives a preliminary diagnosis based on the computed overlaps for each sample.

**Figure 2 diagnostics-12-01968-f002:**
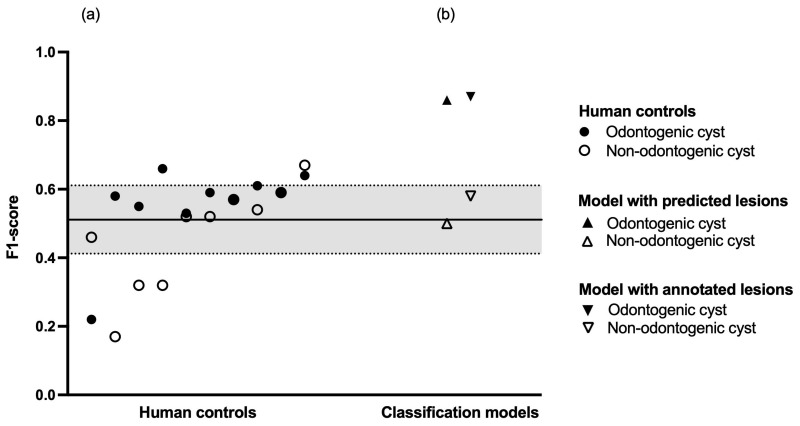
Diagnostic performance. (**a**) Human controls in increasing order of mean F1-score. The black line represents the mean F1-score of the entire human control group, the dashed lines represent plus or minus one standard deviation, respectively. (**b**) Classification models in increasing order of mean F1-score.

**Table 1 diagnostics-12-01968-t001:** Dataset characteristics.

	Vienna	Nijmegen	Total
**Demographics**			
Age, median (IQR), years	50 (39–60)	57 (45–66)	53.5 (41–63)
Female, n (% of total)	391 (43)	115 (35)	506 (41)
Male, n (% of total)	519 (57)	214 (65)	733 (59)
**Diagnosis**			
Cysts, n.f.s., n (%)	215 (23.6)	0 (0)	215 (17.3)
Odontogenic cysts, n (%)	485 (53.3)	102 (31)	587 (47.3)
Non-odontogenic cysts, n (%)	90 (9.9)	139 (42.2)	229 (18.5)
Negative controls, n (%)	120 (13.2)	88 (26.7)	208 (16.8)

IQR, interquartile range.

**Table 2 diagnostics-12-01968-t002:** Classification performance.

	Odontogenic Cyst	Non-Odontogenic Cyst
**With predictions via object detection (IoU ≥ 0.30)**		
Sensitivity (Recall)	0.84	0.56
Specificity	0.56	0.84
PPV (Precision)	0.89	0.45
NPV	0.45	0.89
F1-score	0.86	0.50
**With original annotations**		
Sensitivity (Recall)	0.91	0.51
Specificity	0.51	0.91
PPV (Precision)	0.83	0.68
NPV	0.68	0.83
F1-score	0.87	0.58

IoU, intersection over union; NPV, negative predictive value, PPV, positive predictive value.

**Table 3 diagnostics-12-01968-t003:** Human diagnostic performance.

	Odontogenic Cyst	Non-Odontogenic Cyst	No Cyst
Sensitivity (Recall)	0.70	0.44	0.56
Specificity	0.62	0.95	0.76
PPV (Precision)	0.53	0.58	0.78
NPV	0.83	0.94	0.62
F1-score	0.56	0.45	0.61

NPV, negative predictive value, PPV, positive predictive value.

## Data Availability

Data supporting the findings of the present study are available from the corresponding author upon reasonable request.

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
