# Peer review of "Emulating Clinical Diagnostic Reasoning for Jaw Cysts with Machine Learning"

_diagnostics, 2022, doi:10.3390/diagnostics12081968_

Round 1

Reviewer 1 Report

The authors have presented a paper on cyst detection and classification. The study is overall nicely structured, I have some minor comments.

In introduction, before the aim kindly mention what is unknown about the topic at hand and then present your aim. As there are already studies evaluating cystic lesions based on AI so what does your study aims to add to the already existing literature.

How was the sample size calculated. Kindly add in methodology.

Line 40 “approach to the detection and classification cystic lesions….” “ replace “to” with “for” and add “of” before cystic…”

Remove line 43-46 starting from “To this aim…” it will be in methodology.

Add radiographic definition of odontotogenic and developmental cyst in the methodology with a few radiographic figures of each for proper differentiation. So it would be clear what the human group was looking at. It could be added somewhere near line 166 or upto authors.

No need for line 82-90. Just stick with algorithm you used and rephrase its description accordingly.

Line 99 provide the reasoning for using Adam optimizer.

Line 121 why is testing set so small (10%)?

Provide specialty and experience in years of the dental professionals in human control group. Were they all general dentists?

Were the dentists trained and calibrated, if yes how? Or they just had to guess what the cyst was based on their experience? As experience can also be factor for biased results

Line 177 , 179, table 1 etc provide also male n/%.

In results add a single heading called “Detection and segmentation performance” instead of separate headings.

Significance testing of the reported data is missing in results. Kindly add in statistical analysis the tests done and also add in the results section whether there were any significant differences.

Author Response

Dear Reviewer,

in the name of all coauthors, allow me to express my gratitude for your thorough review of our manuscript as well as the thoughtful comments. In the following, I will point to the specific changes made to our manuscript as a response to your findings. Please note that your comments are quoted verbatim and are in italic, our responses are in bold.

In introduction, before the aim kindly mention what is unknown about the topic at hand and then present your aim. As there are already studies evaluating cystic lesions based on AI so what does your study aims to add to the already existing literature.

We have improved the entire Introduction section. Information requested in this comment has been added to the manuscript at Lines 31 and 44.

How was the sample size calculated. Kindly add in methodology.

Information requested in this comment has been added to the manuscript at Line 71.

Line 40 “approach to the detection and classification cystic lesions….” ” replace “to” with “for” and add “of” before cystic…”

The referenced text has been rephrased according to this comment.

Remove line 43-46 starting from “To this aim…” it will be in methodology.

The referenced text has been removed according to this comment.

Add radiographic definition of odontotogenic and developmental cyst in the methodology with a few radiographic figures of each for proper differentiation. So it would be clear what the human group was looking at. It could be added somewhere near line 166 or upto authors.

Information suggested in this comment has been added to the manuscript at Line 169 as well as Figure A1.

No need for line 82-90. Just stick with algorithm you used and rephrase its description accordingly.

The referenced text has been removed and rephrased according to this comment.

Line 99 provide the reasoning for using Adam optimizer.

Information requested in this comment has been added to the manuscript at Line 95.

Line 121 why is testing set so small (10%)?

Information requested in this comment has been added to the manuscript at Line 119.

Provide specialty and experience in years of the dental professionals in human control group. Were they all general dentists?

Information requested in this comment has been added to the manuscript at Line 163.

Were the dentists trained and calibrated, if yes how? Or they just had to guess what the cyst was based on their experience? As experience can also be factor for biased results

Information requested in this comment has been added to the manuscript at Lines 178 and 279.

Line 177 , 179, table 1 etc provide also male n/%.

Information requested in this comment has been added to the manuscript at Line 187 and Table 1.

In results add a single heading called “Detection and segmentation performance” instead of separate headings.

The referenced text has been removed and rephrased according to this comment.

Significance testing of the reported data is missing in results. Kindly add in statistical analysis the tests done and also add in the results section whether there were any significant differences.

We thank the reviewer for this suggestion, yet respectfully note that since the modeling process has been performed without cross validation, significance testing of our results was not feasible, especially not in the time frame of five days provided for this revision.

We thank you again for your thorough review of our work and hope to have sufficiently addressed your concerns with this revised verson.

The Authors

Reviewer 2 Report

Aauthors discuss deep learning-segmentation based diagnostics of odontogenic and developmental jaw-cysts.

The number of cases is high enough to reach relevant results, the question is how the research and its results are related to clinical work, how can this be used in everyday practice?

In my opinion, the method can only be used only with a thorough clinical examination, it is important to get informaton about vitality, mobility, possible discoloration of the affected and/os neighboring teeth.

Due to all these reasons, this procedure can be used to a limited extent in telemedicine.

I ask the authors to show how the obtained results can be used with safety in the clinical practice.

Author Response

Dear Reviewer,

in the name of all coauthors, allow me to express my gratitude for your review of our manuscript as well as your important comments. In the following, I will respond to the points you raised as well as refer to the specific changes made to our manuscript in light of your comments. Please note that your comments are quoted verbatim and in italics, our responses are in bold.

The number of cases is high enough to reach relevant results, the question is how the research and its results are related to clinical work, how can this be used in everyday practice?

The primary aim of our approach was to — for the first time — pick apart the steps of the human clinical reasoning process and emulate them step by step using machine learning. We believe that the results suggest an adequate approximation of the human clinical reasoning process through our models. In everyday practice, our models can be used to perform a preliminary classification of the suspected cystic lesion with regards to its background: odontogenic or developmental. This is of clinical relevance as it improves the clinician’s confidence whether dental pathology is involved, thus whether endodontic treatment is necessary. Of course, the models are only a supporting tool and should never be used without a thorough clinical examination.

In my opinion, the method can only be used only with a thorough clinical examination, it is important to get informaton about vitality, mobility, possible discoloration of the affected and/os neighboring teeth.Due to all these reasons, this procedure can be used to a limited extent in telemedicine.

We completely agree with you. Information suggested in this comment has been added to the manuscript at Line 290.

I ask the authors to show how the obtained results can be used with safety in the clinical practice.

Naturally, the safety of the patient should be the primary consideration with every new tool, whether diagnostic or therapeutic. Therefore, no diagnosis should be made with our model alone. Importantly, this was already our principle during model development as we only considered histologically confirmed diagnoses for our ground truth. Information requested in this comment has been added to the manuscript at Line 294.

We again thank you for your review and hope to have sufficiently addressed your comments in this revised version of our manuscript.

The Authors